# Pathomechanism of Triangular Fibrocartilage Complex Injuries in Patients with Distal-Radius Fractures: A Magnetic-Resonance Imaging Study

**DOI:** 10.3390/jcm11206168

**Published:** 2022-10-19

**Authors:** Beom-Soo Kim, Chul-Hyun Cho, Kyung-Jae Lee, Si-Wook Lee, Seok-Ho Byun

**Affiliations:** Department of Orthopedic Surgery, Keimyung University Dongsan Hospital, Keimyung University School of Medicine, Daegu 42601, Korea

**Keywords:** distal-radius fracture, triangular fibrocartilage complex injury, magnetic-resonance-imaging study, radial length, open reduction

## Abstract

Injury to the triangular fibrocartilage complex (TFCC) is one of the most common complications following a fracture of the distal radius. In this study, an examination of TFCC injuries in patients with distal-radius fractures was conducted using magnetic-resonance imaging (MRI); the aim of the study was to analyze the prevalence of TFCC injury as well as to suggest acceptable radiologic parameters for use in prediction of the injury pattern. Fifty-eight patients with distal-radius fractures who underwent MRI prior to undergoing open-reduction surgery between April 2020 and July 2021 were included in this study. An analysis of various radiologic parameters, the fracture type, and the MRI classification of TFCC injuries was performed. Radiologic parameters were used in the evaluation of distal radioulnar joint (DRUJ), radial shortening, and the dorsal angularity of the fracture. All of the patients in this study had definite traumatic TFCC injuries. A statistical relationship was observed between the radial length gap between the intact wrist and the injured wrist, which represents relative radial shortening, and the pattern of TFCC injury. In conclusion, the shortening of the distal radius, causing peripheral soft tissue of the ulnar side to become tauter, is highly relevant with regard to the pattern of TFCC injury. However, because no data on the clinical outcome were utilized in this study, it is lacking in clinical perspective. The conduct of further studies on patients’ clinical outcome will be necessary.

## 1. Introduction

Distal-radius fracture (DRF), one of the most common fractures occurring in elderly people, accounts for approximately 18% of fractures in patients older than 65 years [1,2]. Triangular fibrocartilage complex (TFCC) tear, the injury most associated with unstable distal-radius fractures, has been reported in 39% to 84% of cases [2,3]. This concomitant injury may contribute to the development of chronic wrist pain, decreased grip strength, and restricted motion [4].

Better visualization and diagnosis of TFCC injury can be achieved by use of arthroscopic examination; however, it is not a standardized test for use in all patients with distal-radius fractures [5,6]. Although MRI scanning is used for diagnosis of TFCC injuries, MRI testing is not performed routinely in all patients with distal-radius fractures at the time of the injury [7]. Instead, the test is recommended for patients who have symptoms related to TFCC injury after the fracture treatment has ended, which can cause a delay in the treatment of the injury.

Some studies using MRI in patients with DRF have demonstrated the prevalence of TFCC injuries; however, studies on the pathomechanism of TFCC injuries concomitant to DRF have rarely been reported [7]. According to findings from previous studies, radiologic features such as the fracture pattern, the magnitude of displacement, and the presence of an ulnar styloid fracture may be independent predictors of TFCC injuries related to DRF [2,8,9]. This study hypothesized that performing an analysis of radiologic parameters in MRI studies of patients with DRF may foster an understanding of the pathomechanism of TFCC injuries.

The purpose of this study is to a conduct radiographic examination and MRI studies in order to determine the fracture mechanism of distal-radius fracture and the prevalence and the pathomechanism of TFCC injuries concomitant to fracture.

## 2. Materials and Methods

Sixty-three patients underwent surgical management for the treatment of distal-radius fracture in a single fellowship-training hospital between March 2020 and July 2021. Inclusion criteria were patients who underwent open reduction and internal fixation of the fracture. Patients younger than 18 years old and those who had previously occurring arthritis of the wrist or degenerative TFCC injuries on the affected wrist were excluded. Those patients underwent MRI scanning, and five patients who refused the test were excluded, so that 58 patients were finally included in this study (Figure 1).

An analysis of simple radiographic parameters (the radial inclination, the radial length, the distal radioulnar joint (DRUJ) gap, the sagittal/radial transition ratio, the DRUJ gap on the unaffected wrist, and the presence of a distal ulnar fracture) and patterns of TFCC injury in the MRI scan was performed by two orthopedic surgeons using the Palmar classification. Regarding the classification of fractures, the Fernandez classification and the AO/OTA classification were used in defining the mechanism of injury and the fracture pattern, respectively. Other assessments included general demographics and underlying osteoporotic disease.

A standard 4 view x-ray of the injured wrist, and AP and lateral views of the uninjured side were obtained for all patients. Measurement of the DRUJ gap distance was performed on both sides in order to better evaluate widening of the DRUJ. The DRUJ distance was defined as the maximum distance between either the volar or dorsal cortical rim of the sigmoid notch of the radius and the ulnar head. The radial translation ratio was calculated as the fraction of the DRUJ gap distance relative to the radioulnar width of the proximal fracture fragment. On the lateral X-ray, the sagittal translation was defined as the distance between the volar cortex of the radius shaft and the volar cortical margin of the distal fracture fragment. The sagittal translation ratio was calculated as the fraction of the sagittal translation to the AP width of the proximal fracture fragment [2,8].

The articular involvement of the fracture and the presence of an ulnar styloid fracture, which was then classified as a tip, middle, or base fracture each separating 1/3 of the ulnar styloid, was evaluated in this study. In addition, ulnar styloid fracture was classified as type 1,2,3 each corresponding to distal to base where the superficial horizontal fibers of the TFCC are inserted, base fracture and proximal to the base fracture, respectively [9,10].

The radial length was defined as the distance between two lines drawn perpendicular to the long axis of the radius on the AP projection from the apex of the radial styloid and the level of the ulnar aspect of the articular surface. The radial length was measured on the uninjured wrist, and the radial length gap between both sides of the wrist was obtained for the evaluation of the pure radial shortening distance (Figure 2).

An MRI examination of the injured wrist was performed on all patients using a 3.0 T MRI scanner (Magnetom 3.0 T, Siemens, Munich, Germany/Ingenia 3.0 T, Philip, Amsterdam, The Netherlands). Statistical analysis was performed using the SPSS statistical package (Version 22.0; IBM, Armonk, NY, USA). The Chi-square test was used for the evaluation of categorical variables, and the T-Test was used for the evaluation of continuous variables. The level of significance was set as *p* value < 0.05.

## 3. Results

The mean age of the patients was 65.21 years (range: 19–89 years; 15 male and 43 female); 15 men and 43 women were included in the study (Table 1). One patient was injured by a direct hit on the wrist, while other patients were injured by the fallen-onto-outstretched-hand (FOOSH) mechanism. According to the data from AO/OTA classification, 16 patients had A2 fractures; 8 patients had A3 fractures; 4 patients had B3 fractures; and 4, 8, and 12 patients had C1, 2, and 3 fractures, respectively (Table 2). According to the Fernandez classification, 29 patients had type I fractures, and 4, 12, and 13 patients had Type II, III, and V fractures. Twenty-six patients showed widening of the DRUJ gap compared to the unaffected wrist. Associated distal ulnar fractures were detected in 38 patients (65%). All patients in this study had a definite traumatic TFCC injury; 1A (*n* = 5), 1B (*n* = 19), 1C (*n* = 33), and 1D (*n* = 1).

No significant relationship was observed between fracture classification (AO/OTA, Fernandez) and types of TFCC injury (Palmar classification): AO/OTA, Fernandez (Table 3). Intra-articular fracture involvement, and the presence of an ulnar styloid fracture, had no significant effect on the type of TFCC injury: articular involvement (*p* > 0.05) and ulnar styloid fracture (*p* > 0.05). Type lC TFCC injuries were significant in patients with osteoporosis, compared with other groups (*p* < 0.01). Regarding age-related statistics, more Type 1C injuries were observed in significantly older patients than in younger patients (*p* < 0.01).

For the radiologic parameters, except for the radial length gap, there was no significant difference in the type of injury (Table 4). The radial length gap between the intact wrist and the injured wrist showed significant relevance with the pattern of TFCC injury. The increased radial length gap showed relevance with type lC injuries (*p* < 0.05).

## 4. Discussion

According to a previous study, TFCC injury with an exceeding dorsal angulation of 32′ can be expected [10]. However, in this study, all TFCC injuries were detected, and dorsal angulation of more than 32′ was detected in only seven of them; there was no statistical significance with the dorsal angularity of the fracture (*p* > 0.05). In a cadaveric study, the displacement of the intact TFCC complex together with the ulnar styloid base fracture fragment was observed, while TFCC avulsion injuries were detected in patients with ulnar styloid tip fractures [10]. In this study, however, no correlation was observed between ulnar styloid fracture and patterns of TFCC injury type.

The ulnocarpal ligaments (ulnolunate and ulnotriquetral ligament) do not insert onto the ulna but are derived from the anterior part of the TFCC, and they connect the carpus to the ulnar by the palmar portion of the radioulnar ligament at its origin—the fovea. Type lC injury is defined as the distal avulsion of the carpal attachment in TFCC. Findings from this study demonstrated an association of distal avulsion with direct radial shortening, which contradicts the previously held common belief that increased dorsal angularity makes the distal avulsion force stronger. When considering the normal variance of ulnar head positioning, radial length alone in an injured wrist might not represent the degree of radial shortening. The measurement of radial length discrepancy compared with the intact wrist should be performed, and a gap of more than 6.3 mm (SD 4.7) might strongly suggest type lC TFCC injury.

As demonstrated in previous cadevaric studies, ECU subsheath (sECU), an integral part of the TFCC, provides ulnocarpal stability and appears to precede dorsal and palmar injuries. In the Bowstring phenomenon, which explains the rupture of sECU, avulsion injuries of dorsal soft tissues of the TFCC complex are manifested [10,11]. Findings from other studies have demonstrated that the dorsal angulation of distal-radius fractures causes the increased traction of palmar ligaments inserting within the foveal region, making them taut in extension and finally resulting in palmar injuries of the TFCC complex [12]. This explains the mechanism by which dorsal and palmar injuries can co-exist.

Previous studies have demonstrated an association of TFCC injuries with the degree of dorsal or volar angulation of the fracture [13,14]. In a cadaver study, sectioned TFCC resulted in increased dorsal angulation [13]. In this study, TFCC injuries were detected in all patients; however, there was no significant relationship between the degree of dorsal angularity and TFCC injury.

The conclusion of this study is that radial compression and the shortening of the distal radius causes the peripheral tear of the ulnar side, preceding the tear of the palmar ligament. The “dart-throwing motion” in the injury mechanism of the distal-radius fracture has been introduced in order to further explain this concept. The dart-throwing-motion (DTM) plane can be defined as the plane on which the functional oblique motion of the wrist occurs [15,16,17,18]. Geometric anatomical factors, ligament factors, and musculature factors have been used to explain DTM, the functional ROM that extends wrist function with radial deviation (so called radial-extension) and flexes with ulnar deviation (so called ulnar-flexion). Except for one case that had type B TFCC injury—injured caused by a direct blow from a ball—other patients were injured by the fallen-onto-outstretched-hand (FOOSH) mechanism. When considering a dorsal angulated fracture with DTM, the axial compression force might accompany wrist extension and radial deviation not only with wrist extension, and vice versa.

Axial loading and radial deviating force during the injury mechanism of distal-radius fracture causes the shortening of the radial length. Regarding this concept of injury mechanism, the findings from this study demonstrated that radial shortening with a dorsally angulated fracture, regardless of the degree of angulation, increases tension in the soft tissue of the ulnar side, leading to the dorsal or palmar TFCC injury of the ulnar side.

Radial avulsion injury (D1) was detected in only one case, which showed a volar angulated fracture and the greatest increase in radial-length distance (Figure 3). In this case, both distal radioulnar fractures occurred; however, there was greater displacement of the distal ulnar fracture fragment, which was shortened in length, and the distal radius was volar angulated. The patient was driving at the time of injury, with her wrist in a flexed position; as the car came to a sudden stop, she suffered a direct injury from the car handle with her wrist in a flexed position. In this case, considering DTM, the direct axial force that was applied while her wrist was in a state of ulnar flexion caused greater displacement of the distal ulnar fracture. Unlike Colle’s fracture mentioned above, ulnar-deviated and volar-angulating axial force decreases the tension on the palmar soft tissue of the TFCC and makes the detachment of sECU from the ulnar side more difficult, thereby transmitting the axial force to the radial avulsion of the TFCC.

MRI tests have been performed in all patients within 1–2 days after visiting our emergency center. All of the patients took MRI tests within 5 days after injury since some of them visited our clinic 2–3 days after the initial injury. Using the 3.0 T MRI scanner, examiners (two orthopedic specialists) could definitely find acute infiltration or the torn part of the TFCC, which were irrelevant with regard to joint effusion and bone-marrow edema caused by an acute injury. The final reading paper was by a radiologic specialist who also reported a definite traumatic tear in every test. This study shows a higher prevalence of TFCC injury than other studies reported, but because of the relatively poorer sensitivity of the MRI test compared to the arthroscopic examination, the possibility of false-positive results should never be overlooked [6]. Arthroscopic examination facilitates the reduction in intra-articular fracture, and it is a standard study for evaluating concomitant injury involving TFCC tear after DRF [5,6,9]. However, in this study, routine arthroscopy was not performed because patients did not fully consent to having an additional arthroscopic procedure involving general anesthesia (this procedure was not cost-efficient according to the Korean insurance system).

This study has several limitations. First, data on the clinical outcome were not utilized. TFCC injury is the most associated secondary injury after distal-radius fracture; however, because both of these injuries are mostly self-limiting or easily overcome with conservative management, including physical therapy or medication, treatment after healing of the fracture is controversial [19,20]. As demonstrated in this study, most of the patients had concomitant TFCC injuries; however, the overall outcomes after the surgery were not proven. In order to prove the clinical importance of this study, an analysis of patient outcome, such as the clinical score (VAS, Mayo, and DASH); the follow-up data; the period of symptom resolution; and the number of patients receiving additional TFCC management, including surgery, should be performed in the next study. The relatively small number of patients is another limitation of this study. Type D injury was detected in only one case, and the results cannot be supported.

## 5. Conclusions

In conclusion, TFCC injuries were associated with all distal-radius fracture cases in this study, regardless of types of fracture and the presence of ulnar styloid fracture. We concluded that distal radius shortening, resulting in the avulsion of TFCC ligaments, is the predominant parameter affecting the TFCC injury pattern.

## Figures and Tables

**Figure 1 jcm-11-06168-f001:**
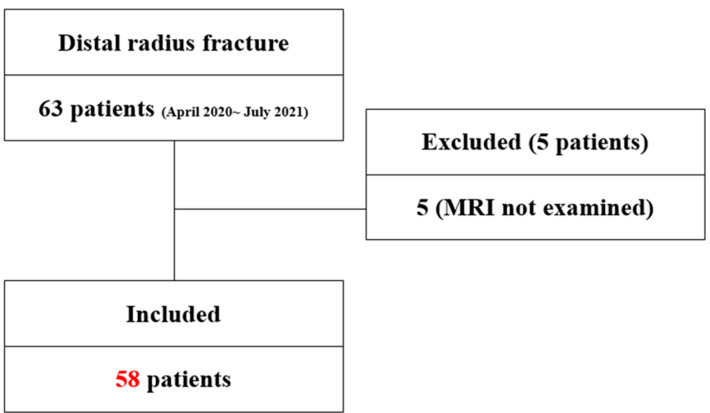
Patient’s flow chart. Patient profiles and the groups included in the study. MRI, magnetic-resonance imaging.

**Figure 2 jcm-11-06168-f002:**
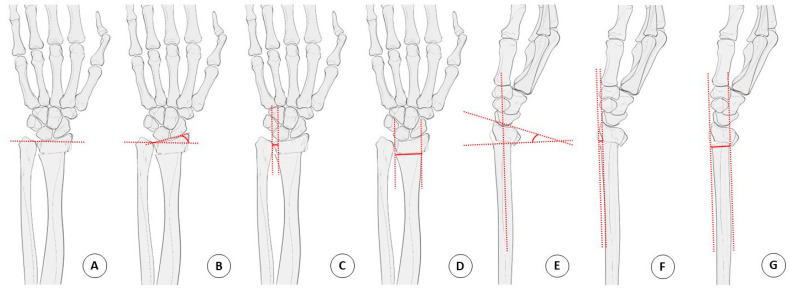
Radiologic parameter measurement technique. (**A**) Radial length. (**B**) Radial inclination. (**C**) DRUJ distance. (**D**) Fracture site width. (**E**) Dorsal angulation. (**F**) Sagittal translation. (**G**) Anteroposterior width. Radial translation ratio = C/D, Sagittal translation ratio = F/G. DRUJ, distal radioulnar joint.

**Figure 3 jcm-11-06168-f003:**
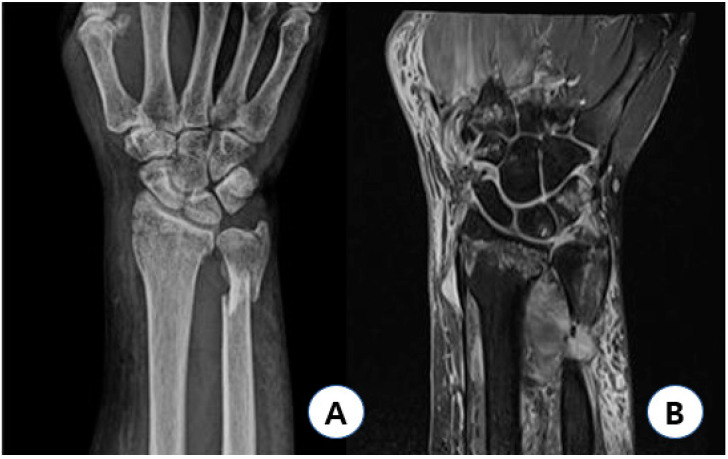
Type 1D injury case. Sixty-three year-old female patient injured with her wrist flexed had AO/OTA type A2 fracture with 15′ volar angulation, showing 5 mm increased radial length gap (**A**), and had type 1D TFCC injury with relatively preserved peripheral and distal portion of TFCC (**B**). AO: arbeitsgemeinschaft fur osteosynthesfragen; OTA: orthopedic trauma association; and TFCC: triangular fibrocartilage complex.

**Table 1 jcm-11-06168-t001:** Demographic data.

Index		TFCC Pattern Relevance (*p* Value)
Index age (average)	65.21	>0.05
Gender (male:female ratio)	8:2	>0.05
BMI (average)	24.05	>0.05
Osteoporosis (%)	37.9	>0.05

BMI: body mass index; TFCC: triangular fibrocartilage complex.

**Table 2 jcm-11-06168-t002:** TFCC injury pattern.

TFCC Injury Pattern (Palmar Classification)	N
1A	5
1B	19
1C	33
1D	1

N: number; TFCC: triangular fibrocartilage complex.

**Table 3 jcm-11-06168-t003:** Fracture classification with TFCC injury pattern.

	Palmar Classification				
	1A	1B	1C	1D	Total
AO/OTA classification					
A2	2	5	9	1	17
A3	0	5	6	0	11
B3	1	2	1	0	4
C1	1	1	2	0	4
C2	0	4	6	0	10
C3	1	2	9	0	12
Total	5	19	33	1	58
Fernandez classification					
1	2	11	15	1	29
2	1	2	1	0	4
3	1	3	8	0	12
5	1	3	9	0	13
Total	5	19	33	1	58

Fracture classification (AO/OTA, Fernandez) had no statistical significance with TFCC injury. (*p* > 0.05/*p* > 0.05 respectively). AO, arbeitsgemeinschaft fur osteosynthesfragen; OTA, orthopedic trauma association; and TFCC, triangular fibrocartilage complex.

**Table 4 jcm-11-06168-t004:** Radiologic parameter with TFCC injury pattern.

	Palmar Classification				
Radiologic Parameter	1A	1B	1C	1D	*p*-Value
DRUJ gap	−0.014	0.28	0.25	0.07	>0.05
Radial length	6.69	5.25	3.14	11.9	>0.05
Radial length gap	3.53	5.04	6.27	−5.16	<0.05 *
Radial inclination	18.06	14.026	13.982	23.9	>0.05
Dorsal angulation	−2.02	5.13	7.84	−15.9	>0.05

Radiologic parameter, except radial length gap had no significant relevance with TFCC injury pattern. DRUJ, distal radioulnar joint; TFCC, triangular fibrocartilage complex; and *, statistically significant.

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
