# Peer review of "Pathomechanism of Triangular Fibrocartilage Complex Injuries in Patients with Distal-Radius Fractures: A Magnetic-Resonance Imaging Study"

_jcm, 2022, doi:10.3390/jcm11206168_

Round 1
Reviewer 1 Report
The authors have performed an MRI study of a cohort of displaced distal radius fracture prior to manipulative reduction and further treatment. The MRI observations confirm that the ulnar side of the wrist, particularly the TFCC, are part of the injury and observed that with shortening of the radius as compared to the uninjured limb, the soft tissues on the ulnar side become more taut
This study confirms conclusively that the soft tissues on the ulnar side of the wrist are part of the traumatic injury. The documentation is clear and statistically relevant
While the study lacks any ability to correlate the MRI findings with either any impact on treatment decision making nor on the ultimate outcome of the injury and treatment, it still provides documentation of the ulnar side of the wrist as a relevant part of the injury
What certainly be worth wile would be to follow this with a longitudinal study extending to treatment and clinical outcome
Author Response
Dear. Reviewer 1
Thank you for your sincere response.
According to your comments, we did our best to revise our paper. We believe that this process makes our study more valuable. Our answers to your comments are below.
<Response to Reviewer 1 Comments>
Reviewer 1
The authors have performed an MRI study of a cohort of displaced distal radius fracture prior to manipulative reduction and further treatment. The MRI observations confirm that the ulnar side of the wrist, particularly the TFCC, are part of the injury and observed that with shortening of the radius as compared to the uninjured limb, the soft tissues on the ulnar side become more taut
This study confirms conclusively that the soft tissues on the ulnar side of the wrist are part of the traumatic injury. The documentation is clear and statistically relevant
While the study lacks any ability to correlate the MRI findings with either any impact on treatment decision making nor on the ultimate outcome of the injury and treatment, it still provides documentation of the ulnar side of the wrist as a relevant part of the injury
What certainly be worth wile would be to follow this with a longitudinal study extending to treatment and clinical outcome
-> Thank you for your sincere response. We are dealing with this study as a preliminary step for further evaluating patients’ clinical outcome and treating them based on our data. We are still gathering patients clinical score in outpatient clinic, following up their ulnar side pain and if adequate number of follow up studies are accumulated, we will soon start up next study. Next studies might include treatment options for patients who suffers from long term ulnar side wrist pain after treating with distal radius fractures and so on.

Reviewer 2 Report
1. The parameter of ulnar variance didn't be mentioned. I think this data can support your conclusion.
2. I supposed that we can know the "pathomechanism" between the fracture pattern of distal radius and TFCC classification diagnosed by MRI in your study. But I think less information was shown in your discussion. DTM is the only mechanism you mentioned in the content. I think it's not enough and propriate in your study. In your research design, you can only explain the relationship of plain film and MRI. However, more statistic results were nonsignificant.
3. I cannot find some important contribution in your Table. The information in your table about TFCC classification and injury mechanism of distal radius fracture was not enough.
Author Response
Dear. Reviewer 2
Thank you for your sincere response.According to your comments, we did our best to revise our paper. We believe that this process makes our study more valuable. Our answers to your comments are below.
<Response to Reviewer 2 Comments>
- The parameter of ulnar variance didn't be mentioned. I think this data can support your conclusion.
-> Thanks for your comment. Since we measured radial length between end of radial styloid and upper margin of ulnar articular surface, we proposed that radial length and radial length gap can represent relative difference of ulnar variance between injured and un-injured wrist. However according to your precious comment, ulnar variance measurement should be added up in next study.
- I supposed that we can know the "pathomechanism" between the fracture pattern of distal radius and TFCC classification diagnosed by MRI in your study. But I think less information was shown in your discussion. DTM is the only mechanism you mentioned in the content. I think it's not enough and propriate in your study. In your research design, you can only explain the relationship of plain film and MRI. However, more statistic results were nonsignificant.
-> Thanks for your comment. Through our statistics, radial shortening measured on plain film was related with peripheral TFCC tear. There were no other studies correlating functional ROM or “Dart-throwing motion” with injury mechanism of wrist fractures. However, in most cases, patients are injured with FOOSH mechanism with wrist dorsally flexed and because of its “radial -extension” mechanism, we suggest the idea that giving more extension and tension to ulnar side of the wrist, peripheral tears might be more prevalent in such injury mechanism. We added this in the discussion section.
- I cannot find some important contribution in your Table. The information in your table about TFCC classification and injury mechanism of distal radius fracture was not enough.
-> Thanks for your comment. Except one case having type B TFCC injury which was injured by a direct blow by a ball, all of the patients were injured by FOOSH injury mechanism. We added in the result section.

Reviewer 3 Report
This study investigated that prevalence of triangular fibrocartilage complex (TFCC) injury associated with distal radius fracture (DRF) and the relevance between radiological parameters and TFCC injury patterns. The authors concluded that increased radial shortening represents significant relevance with type 1C injuries.
This study is of potential interest as the authors described relation between X-ray parameters and TFCC injury patterns. The pathology of TFCC injury which is concomitant with DRF is still unclear because it is often difficult to perform MRI for all patients in the acute phase after fracture as the authors stated. However, several points as indicated below need to be addressed by authors to improve the quality of the article.
The previous studies performed at conventional 1.5-T MRI ranged from sensitivities of 52-100% and specificities of 50-100% for detection of TFCC tears compared with arthroscopy. It is also known that peripheral ulnar-sided TFCC tears can be more difficult to detect on conventional MRI (not arthrography). Some studies reported that improved depiction with 3-T MRI was associated with higher diagnostic accuracy in the detection of TFCC tears. However, despite the usefulness of the imaging studies, arthroscopy remains the gold standard for evaluating TFCC pathology. In the acute phase, the diagnosis of TFCC injury and determination of the Palmer type may be inaccurate. With regard to this background, I have made some comments.
Therefore, I recommend “minor revise” including following some suggestions.
These are itemized below:
1. Materials and Methods. Please indicate how many days after the injury MRI was performed because TFCC injury might not be properly evaluated immediately after the fracture due to the presence of hematoma and inflammation, etc. Please provide the literature in the discussion section, if any, on how long after the acute phase of fracture the TFCC injuries can be properly assessed by MRI.
2. The manuscript stated that the prevalence of TFCC injury associated with distal radius fractures was 100%, which seems much higher compared to the previous reports. What would you think brought this divergence? Could you mention about that in the discussion section?
3. A certain arthroscopic study reported the highest incidence of Palmer 1D tears in patients with DRF while there was only one case of 1D injury in the present study. These discrepancies are possibly due to diagnostic inaccuracies of MRI. In this study, was arthroscopic diagnosis performed at the time of surgery? If not, please write it as limitation.
4. Table 1. What “8:2” in “Gender” line means? Should it be 2:8 or 3:7 as gender ratio?
5. Line 140. Please add the reference of “In a cadaveric study”.
6. Line 212. “in only case” should be “in only one case”.
Author Response
Dear. Reviewer 3
Thank you for your sincere response.According to your comments, we did our best to revise our paper. We believe that this process makes our study more valuable. Our answers to your comments are below.
<Response to Reviewer 3 Comments>
Reviewer 3
This study investigated that prevalence of triangular fibrocartilage complex (TFCC) injury associated with distal radius fracture (DRF) and the relevance between radiological parameters and TFCC injury patterns. The authors concluded that increased radial shortening represents significant relevance with type 1C injuries.
This study is of potential interest as the authors described relation between X-ray parameters and TFCC injury patterns. The pathology of TFCC injury which is concomitant with DRF is still unclear because it is often difficult to perform MRI for all patients in the acute phase after fracture as the authors stated. However, several points as indicated below need to be addressed by authors to improve the quality of the article.
The previous studies performed at conventional 1.5-T MRI ranged from sensitivities of 52-100% and specificities of 50-100% for detection of TFCC tears compared with arthroscopy. It is also known that peripheral ulnar-sided TFCC tears can be more difficult to detect on conventional MRI (not arthrography). Some studies reported that improved depiction with 3-T MRI was associated with higher diagnostic accuracy in the detection of TFCC tears. However, despite the usefulness of the imaging studies, arthroscopy remains the gold standard for evaluating TFCC pathology. In the acute phase, the diagnosis of TFCC injury and determination of the Palmer type may be inaccurate. With regard to this background, I have made some comments.
Therefore, I recommend “minor revise” including following some suggestions.
These are itemized below:
- Materials and Methods. Please indicate how many days after the injury MRI was performed because TFCC injury might not be properly evaluated immediately after the fracture due to the presence of hematoma and inflammation, etc. Please provide the literature in the discussion section, if any, on how long after the acute phase of fracture the TFCC injuries can be properly assessed by MRI.
-> Thanks for your comment. We performed every MRI tests within 1-2 days after visiting our emergency center. Some patients visited our clinic via another clinic after 2-3 days of injury. So we can say that most patients had MRI study within at least 5 days of injury. However in our 3.0T MRI scan, examiners (2 orthopedic specialists) could definitely find acute infiltration and torn site of TFCC irrelevant to joint effusion and bone marrow edema. Final reading paper by radiologic specialists also reported definite traumatic tear in the test. Since recent studies report poor sensitivity compared to arthroscopic examination, we added your precious comment in our discussion.
- The manuscript stated that the prevalence of TFCC injury associated with distal radius fractures was 100%, which seems much higher compared to the previous reports. What would you think brought this divergence? Could you mention about that in the discussion section?
-> Thanks for your comment. We were also surprised about our results. However, even though we could find traumatic tear in MRI studies, as you’ve mentioned in your 1st comment, we should also think about false positive results derived from limitation of MRI studies. We added your precious comment in our discussion.
- A certain arthroscopic study reported the highest incidence of Palmer 1D tears in patients with DRF while there was only one case of 1D injury in the present study. These discrepancies are possibly due to diagnostic inaccuracies of MRI. In this study, was arthroscopic diagnosis performed at the time of surgery? If not, please write it as limitation.
-> Thanks for your comment. Recent studies suggest that arthroscopic examination is a gold standard study for TFCC injuries. However, in Korean medical insurance system where government regulates medical practices and medical costs doesn’t sufficiently compensates its’ practices, arthroscopic examination concomitant to open reduction of distal radius fracture is highly cost in-efficient and patients don’t usually accept with them. We had full consent on having MRI studies and it can be performed before the surgery which is time efficient. We added your precious comment in our limitation.
- Table 1. What “8:2” in “Gender” line means? Should it be 2:8 or 3:7 as gender ratio?
-> Thanks for your comment. “8:2” represented ratio between male and female patient. We added in our Table 1.
- Line 140. Please add the reference of “In a cadaveric study”.
-> Thanks for your comment. We added the reference in the context.
- Line 212. “in only case” should be “in only one case”.
-> Thanks for your comment. We revised that error in our context

Round 2
Reviewer 2 Report
I accept your detailed explanation. I am appreciated to your team's effort.
Detectable Parameter from wrist X-ray --- Mechanism of injury --- TFCC injury pattern. The correlation of these three roles is very interesting. More researches are needed.